# Age and Sex-Related Differences in the Tongue Pressure Generated during Maximum Isometric and Swallowing Tasks by Healthy Chinese Adults

**DOI:** 10.3390/ijerph18105452

**Published:** 2021-05-20

**Authors:** Wen-Yu Lin, Yu-Mei Chen, Kuen-Ming Wu, Pei-Kai Chen, Yueh-Juen Hwu

**Affiliations:** 1Nuclear Medicine Department, Jen-Ai Hospital, Taichung 41265, Taiwan; numedicine@mail.jah.org.tw; 2Endocrinology Division, Department of Internal Medicine, Jen-Ai Hospital, Taichung 41265, Taiwan; 3Department of Social Services, Jen-Ai Hospital, Taichung 41265, Taiwan; cym728@gmail.com; 4The Department of Adult and Continuing Education, National Chung Cheng University, Taoyuan 33223, Taiwan; 5Internal Medicine Department, Jen-Ai Hospital, Taichung 41265, Taiwan; kuenming_wu@yahoo.com.tw; 6Neurology Department, Jen-Ai Hospital, Taichung 41265, Taiwan; med@mail.jah.org.tw; 7College of Nursing, Central Taiwan University of Science and Technology, Taichung 40601, Taiwan

**Keywords:** tongue pressure, maximum isometric pressure (MIP), swallowing pressure (SP), presbyphagia

## Abstract

The aims of this prospective observational study were to investigate age, sex, and factors related to the tongue pressure generated. A correlational research design was used. A total of 150 Chinese people who had a normal swallowing condition were enrolled by convenience sampling. Pressure was measured for each participant during maximum isometric press tasks, as well as for saliva and water swallows (5 mL) at the anterior and posterior tongue. The results illustrated that age has an impact on anterior tongue pressure (*r* = −0.22), posterior tongue pressure (*r* = −0.26); however, it does not have an impact on the swallowing pressure (SP) of the tongue. Sex differences were noted; males demonstrated a greater strength of the anterior tongue. There was a significant correlation between BMI and the maximum isometric pressure of the anterior tongue (MIP_ant_). The pressures between anterior and posterior tongue were not significantly different in the maximum isometric or swallowing tasks. There were significant differences among the maximum isometric pressure (MIP), saliva swallowing pressure, and water swallowing pressure. The MIP generated was greater than the pressure in the swallowing tasks for the younger groups of both sexes. The study supplement the exploration of age-and-sex related differences and the interaction of sex and age in tongue pressure.

## 1. Introduction

The tongue is responsible for preparing, forming, manipulating, and transferring boluses to the pharyngeal cavity, as well as eliciting the pharyngeal reflex to propel the bolus downwards. Normal tongue strength is necessary for swallowing. One of the ways to examine the function of the tongue muscle is to measure tongue pressure during maximum isometric and swallowing tasks [1].

Age, sex, and the areas of the tongue influence the maximum isometric pressure (MIP) in healthy adults [2,3,4]. Previous studies documented the decline of the MIP of the tongue with age, which can negatively impact swallowing function [4,5,6,7,8]. A significant age-related decline in tongue muscle performance is commonly associated with presbyphagia. Sarcopenia is one of the leading causes of presbyphagia [9].

The research findings on the effects of age and sex on tongue strength have not been consistent. Adams et al. [10] conducted a systematic review and investigated the impacts of sex, age, and the areas of tongue during MIP generation [10]. Their results indicated that the mean peak pressure values ranged from 43–78 kPa for both the anterior and posterior tongue in healthy adults, although the average MIP_ant_ is higher than MIP_post_. Meta-analyses further revealed significantly higher MIP in male subjects compared with female subjects, and higher tongue pressure in adults aged <60 years compared with older adult subjects. Tongue pressure is usually lower in females and older adults [10]. In contrast, Vitorino [11] found that there were no significant differences across different ages or between the sexes. 

Tongue pressure generated during swallows is considerably lower than the MIP that healthy individuals are capable of producing [3,12]. Therefore, swallowing is considered a submaximal-force lingual task. Bolus viscosity has been identified as a factor affecting lingual-palatal pressure generation during swallows [13]. Anterior saliva swallowing pressure (SSP_ant_) is higher than posterior saliva swallowing pressure (SSP_post_), but swallowing pressure varies with liquid consistency [3,12]. However, solid boluses require higher posterior pressure generation compared with anterior pressure, presumably for bolus propulsion into the pharynx [2]. Individuals with MIP measurements < 40 kPa report greater difficulty in swallowing liquids of thick consistency [14].

Currently, few research studies have reported tongue pressure and related factors in Chinese people [6,15]. Thus, healthcare providers require a comparative basis for the identification of altered tongue pressure. Furthermore, additional research that examines differences in tongue pressure across the age continuum conducted with stratified samples is necessary for comparison with prior research studies to investigate the similarities and differences.

This article reports the results of a substudy of the norm data, which investigated the tongue pressure among Chinese people in Taiwan [16]. The aim of this particular substudy is to determine the associations of age, sex, BMI, different pressure generating tasks, and areas of the tongue with the pressure that is generated.

Investigators hypothesized that younger adults would exhibit higher tongue pressure than their older counterparts (H_1_). Furthermore, males were expected to be able to generate higher tongue pressure than females (H_2_). In addition, there would be differences in tongue pressure for various tasks (H_3_). The final hypothesis was that the interaction of sex and age may contribute to variations in tongue pressure [15,17] (H_4_). The influences of BMI and the areas of the tongue were also investigated. The current study can supplement the exploration of age-and-sex-related differences and the interaction of sex and age in tongue pressure. Using different tasks measuring tongue pressure provides normative data for healthcare providers to use for the identification of individuals with swallowing difficulties.

## 2. Methods

A correlational research design was conducted to investigate the associations among age, sex, BMI, MIP, and swallowing pressure (SP) generated by the anterior and posterior tongue.

### 2.1. Participants 

To investigate differences in the age-related decline in tongue pressure, the participants consisted of 150 Chinese people recruited by convenience sampling. Of the 150 participants, 49 were male and 101 were female (age range: 20–79 years; mean ± standard deviation [SD]: 36.1 ± 14.9 years). There were no participants with oral motor disease. More than half of the participants (54.0%) had an abnormal body mass index value (<18.5 or >24.0 kg/m^2^). The included participants reported good lip, teeth, tongue, palate, and chewing function. An in-depth interview revealed that none of the participants had a history or current diagnosis of neurological disease, head and neck surgery or injury, or other oral conditions that may impact lingual function [16].

All participants were informed of the objectives of the study and provided written consent. The study protocol was approved by the Institutional Review Board of the Jin-Ai Hospital, Taichung, Taiwan (No.109-69). All procedures were conducted in accordance with the ethical standards of the responsible committees on human experimentation and the Helsinki Declaration of 1964 and its later versions.

### 2.2. Lingual Strength Evaluation

#### 2.2.1. Maximum Isometric Pressure (MIP)

The Iowa Oral Performance Instrument (IOPI) is a portable, hand-held device that consists of a pressure transducer and an amplifier that displays (in kPa) the pressure exerted on an air-filled bulb. The air-filled bulb (approximately 3.5 cm long and 4.5 cm in diameter) is placed between the tongue and palate. To ensure accurate measurement, calibration of the IOPI was conducted once a week according to the IOPI manual. Participants were instructed to press the bulb against the hard palate for 2 s after it was placed immediately behind the alveolar ridge (anterior tongue) or aligned with the first molar (posterior tongue) to measure the tongue pressure. The MIP_ant_ and MIP_post_ were obtained first.

The IOPI is the most frequently used device for collecting tongue pressure data [13]. This instrument is the international standard method for research and used in clinical practice; it has high inter-rater and intra-rater reliability [4,11].

#### 2.2.2. Swallowing Pressure (SP)

SP was defined as the non-effortful swallowing pressure across three consecutive trials for saliva swallowing and thin liquid (commercially bottled water) swallowing. The bulb was placed in the specified anterior or posterior position used in the earlier MIP measurements. The bolus was offered by the investigator in a cup (5 mL) and swallowed by the participant with the tongue bulb in the lingual regions in a comfortable manner. 

SSP_ant_ and SSP_post_ were assessed first, followed by WSP_ant_ and WSP_post_. A rest period of 5 min was provided between measures to avoid fatigue. The MIP and SP tasks were performed in the same order by all participants.

### 2.3. Data Collection

Demonstration and practice with verbal encouragement from a trained research assistant were completed prior to the initiation of data collection. Participants were instructed to perform three trials of each task at a comfortable pace (i.e., with 10 s of rest between trials). The peak pressure recorded during the three trials represented the MIP or SP of tongue pressure.

The research assistant monitored the bulb position throughout each set of task repetitions. If bulb movement was suspected, the participants were asked to open their mouth so that bulb placement could be confirmed and corrected. Repeated measures analysis of covariance was used and no statistically significant differences in tongue pressure were found across the three trials of the maximum isometric and swallowing tasks. 

### 2.4. Statistical Analysis 

All tongue pressures were continuous data and independent between subjects, which met the assumptions for parametric analyses. Analysis of variance (ANOVA) was performed to determine whether the variable of tongue pressure differed significantly based on age, sex, BMI, and tasks using the Tukey honestly significant difference procedure for pairwise comparisons. Paired-t tests were used to analyze the areas of tongue with generated pressure. Age (in decades) was used to compare between-groups differences; the exact age was used to evaluate correlations between strength parameters. In addition, linear regression was used to examine the correlation between MIP and parameters of participants. The effect size was calculated using η^2^. Values < 0.06, 0.06–0.14, and >0.14 indicated small, medium, and large effect sizes, respectively [18].

In addition, mixed and repeated measures factorial ANOVA was performed using the MIP and SP generated based on the independent variables of age group and sex. A *p* value < 0.05 denoted a statistically significant difference.

## 3. Results

### 3.1. Age-Related Differences

The age decade was used when comparing between-groups differences. There were few participants older than 60 years (n = 10) and they were put into one group for practical reasons. There were no statistically significant differences for any outcome variables among the five age groups, except the MIP_post_ (Table 1). The post hoc analysis indicated that participants aged 20–39 years had higher MIP_post_ (56.15 ± 11.85) than those aged >60 years (42.50 ± 8.70). The effect size (calculated using η^2^) was medium (0.10). The Pearson’s *r* correlation coefficients between age and tongue pressure variables were determined, revealing that age has an impact on the MIP of the tongue (anterior tongue pressure *r* = −0.22; posterior tongue pressure *r* = −0.26); however, it does not have an impact on the SP of the tongue. 

### 3.2. Sex-Related Differences

Sex had a statistically significant effect only on MIP_ant_. Male participants demonstrated significantly higher MIP_ant_ than female participants. The effect size (calculated using η^2^) was medium (0.07) (Table 2).

### 3.3. BMI-Related Differences

The BMI parameter was calculated as participants’ body weight divided by their body height squared (kg/m^2^) and dichotomized into a categorical variable (normal: 18.5–24.0 kg/m^2^; abnormal: <18.5 kg/m^2^ or >24.0 kg/m^2^). Further analysis of BMI-associated differences in lingual parameters using one-way ANOVA revealed nonsignificant differences for all parameters (Appendix A Table A1). The Pearson’s *r* correlation coefficients between BMI and tongue pressure parameters were calculated, revealing a statistically significant difference only in MIP_ant_ (*r* = 0.22) (Appendix A Table A2).

Linear regression analysis between MIP and parameters of participants, such as age, sex, and BMI were presented in Appendix A Table A3. Age-related reductions in tongue pressure were significant in MIP_ant_ and MIP_post_.

### 3.4. Areas of Tongue–Related Differences

A paired-samples *t*-test was conducted to evaluate the differences in MIP and SP between the anterior and posterior tongue regions. All mean values for the anterior tongue were higher than the means for the posterior tongue, except for SSP. There were no statistically significant differences between the anterior and posterior tongue pressure for either the MIP or SP (Table 3).

### 3.5. Task-Related Differences

The differences between the pressure generating tasks were assessed using one-way ANOVA to compare the group means of the three procedures (MIP, SSP, and WSP) and the two tongue areas. Statistically significant differences were found for all measurements (Table 4). The magnitude of tongue pressure decreased in a stepwise manner from MIP to SSP to WSP. A post hoc test demonstrated a tendency for a significant difference to be found between any two groups. All effect sizes (η^2^) were medium (i.e., 0.08 and 0.10 for the anterior and posterior tongue). The percentages of the MIP employed during the swallowing of saliva and water are 85% and 80% for the anterior tongue, and 90% and 81% for the posterior tongue, respectively (Table 4).

### 3.6. Isometric vs. Swallowing Task by Age Group and Sex

Participants were stratified by age into young (20–39 years), middle-aged (40–59 years), and old (≥60 years) groups [19]. Mixed and repeated measures factorial ANOVA was performed using the MIP and SP generated dependent on the different age groups and sexes. For the anterior tongue, MIP was higher than the SSP and WSP for male participants in the young and middle-aged groups (*p* < 0.05). For the posterior tongue, this tendency was observed only in the young group (*p* < 0.05) (Figure 1a,b). For female participants, the MIP was higher than the pressure in any of the swallowing tasks for the young and middle-aged groups (*p* < 0.05) for both the anterior and posterior tongue (Figure 1c,d).

## 4. Discussion

Age-related reductions in tongue pressure were clearly observed in MIP, both for the anterior and posterior tongue, and particularly for individuals aged ≥60 years. This finding is in line with the currently available literature [4,10,17,20,21]. The age-related decreases in tongue pressure may be indicative of sarcopenia of the tongue. Sarcopenia is a normal age-related loss of skeletal muscle mass and strength [22], which has also been observed in lingual musculature [5,10]. Additionally, excessive connective tissue and fatty cells progressively accumulate in the tongue with age [23]. These non-muscular tissues may cause age-related reductions in orofacial muscle tone [1,24]. The regression of MIP_ant_ (−0.22) and MIP_post_ (−0.26) as a consequence of age was significant and lower than values reported in some previous studies [4,5], but larger than the values reported by Clark and Solomon (−0.12, −0.14) [16]. These differences may originate from the racial variation.

We expected that older adults would have reduced maximum isometric tongue pressure but not reduced pressure during swallowing relative to the younger groups. According to the one-way ANOVA, this prediction held true for MIP_post_. It may be due to the different muscle fibers in anterior and posterior tongue. Post hoc analysis indicated that participants aged 20–39 years had higher MIP_post_ than those aged > 60 years. The maximum tongue pressure we measured averaged 7–10 kPa lower in the oldest participant group compared with the young group, which is in line with other studies [5,10,17]. The effect of age on tongue SP was nonsignificant. SP remains stable throughout the majority of life, supporting previous findings [4,25,26]. However, in contrast, one Korean study demonstrated that the tongue pressure used during swallowing was statistically significantly higher for older adults than for younger adults [27]. The reason for this effect may be the overall decrease in orofacial muscle strength and tone in older adults. Hence, older adults need to use greater SP to compensate for the reduction [27].

Prandini et al. [26] recruited 51 Brazilian volunteers aged 18–28 years to measure tongue pressure during specific tasks, e.g., tongue elevation, endurance, and swallowing. The results showed that sex had no influence on these tasks. Clark and Solomon [17] investigated the different actions of the tongue (e.g., elevation, protrusion, and lateralization) and also found no statistically significant difference between males and females. In the current study, males showed higher MIP_ant_ than females, in agreement with the majority of previous studies [4,7,8,10]. The mean maximum isometric tongue pressure of males was 3.6–7.8 kPa higher than that of females. This result is also similar to the findings of previous studies showing that mean tongue elevation pressure was greater in males than in females (range: 4–10 kPa) [8,25]. 

Generally speaking, males have greater muscle mass, resulting in increased pressure. Given that the lingual strength reserve becomes more important with age, males appear to be able to generate greater tongue pressure; hence, they may possess a greater resilience to the effects of presbyphagia or dysphagia [25]. The results of this study show that males have a statistically significantly higher MIP_ant_ than females, but not MIP_post_. Hence, the present study partially confirmed the concept that sex-related differences may originate from anatomical differences between the sexes.

A previous study reported a correlation between anterior tongue pressure and BMI [1], which is in line with the results of our study. BMI is closely related to sarcopenia. Muscular atrophy is more often observed in fast-twitch muscles than in slow-twitch muscles of patients with sarcopenia [28]. Anterior tongue muscles primarily contain fast-twitch muscles that might be easily influenced by aged-related sarcopenia [29]. However, tongue strengthening exercises can delay the atrophy of fast-twitch muscle fibers that occurs with aging [15,29].

When measuring the MIP and SP of the tongue, the bulb was positioned behind the central incisors or aligned with the first molars for the anterior tongue or posterior tongue, respectively. Participants were asked to squeeze the bulb against the hard palate. When the position of the bulb is near the pharynx, it may elicit an uncomfortable feeling or interfere with swallowing. Residents in long-term care facilities may not tolerate the posterior placement of the bulb. Peladeau-Pigeon and Steele [30] claimed that swallowing with an air-filled bulb in the mouth is unnatural and could influence tongue pressure amplitudes. Since there were no statistically significant differences in MIP or SP between the anterior and posterior tongue, the former region may be more appropriate for conducting measurements in clinical practice.

Interestingly, SPs in this study were lower; on average, than the MIP that could be generated; swallowing saliva resulted in 85–90% of the values obtained on the MIP task, while swallowing water resulted in 80–81% of the MIP task values. Peladeau-Pigeon and Steele [30] analyzed the swallowing tasks of 84 healthy participants and found that the mean amplitudes of the saliva swallows ranged from 70–81% of the MIP. Youmans et al. [5] investigated the SP required for different liquids and reported 50–60% of the values obtained for the MIP task.

The results of this exploratory study confirm previous findings that MIP is greater than SSP and WSP, while SSP is greater than WSP [5,30]. Additionally, the present study finds that the above phenomenon is prominent for both the anterior and posterior tongue in the young group (20–39 years) for both sexes. It is likely that the bolus distributes the pressure generated by the tongue musculature over the palate in a pattern determined by the consistency of the bolus leading to varying pressure patterns [19]. Overall, we have shown that the tongue pressure employed in different tasks gradually declines with increasing age for both sexes, except in the older adult group (aged ≥60 years) (Figure 1). It might originate from the fewer participants in the older adult group and thus limit the outward differences among groups.

## 5. Limitations and Suggestions

Although the present study reveals important reference variables in tongue pressure for maximum isometric and swallowing tasks, some limitations exist. Firstly, the presence of fewer subjects in the oldest age category, and fewer males than females may elicit selection bias. Therefore, a study involving a sufficiently diverse group of participants to capture age- and sex-related changes is warranted to verify the results of the present study. Secondly, this is the first study measuring the MIP and SP of the tongue in Chinese individuals. Replication of the study is needed to confirm the present results. In addition, only healthy Chinese people were recruited in the present study. Therefore, it is not possible to draw conclusions regarding possible differences in tongue pressure compared with other populations. Thirdly, performing an isometric task with maximum effort may carry over into the use of effort during saliva and water swallowing tasks. It would be possible to eliminate this confounding factor by using a random order in the measuring procedure in the future. Furthermore, apart from age and sex, there are numerous other possible factors that can influence tongue pressure, such as dentures, nutrition, and race. These important parameters should be considered in future research. Finally, the interplay of multiple factors affecting aging lingual tissue could be responsible for the observed decrease in tissue strength and must be considered as a potential confounding factor for the generation of normative data. 

## 6. Conclusions

There has not been an investigation on MIP, SSP, and WSP data of the tongue in a Chinese population that varied across age and sex. This study provides the first Chinese data on tongue pressure generating tasks. The results illustrated that age has an impact on anterior tongue pressure (*r* = −0.22), posterior tongue pressure (*r* = −0.26); however, it does not have an impact on the swallowing pressure (SP) of the tongue. Advanced age and female sex appeared to be associated with lower MIP. There were no significant differences between the anterior and posterior tongue pressure for either MIP or SP. There are significant differences in tongue pressure among MIP, SSP, and WSP. The MIP is greater than the pressure generated by either swallowing task for the young groups of both sexes. The MIP, SSP, and WSP data of the tongue help in the comprehension of swallowing physiology, and will thus contribute to therapeutic planning for individuals with dysphagia. The current study expands the dataset describing factors related to tongue pressure measures, especially the interaction of sex and age in tongue pressure. 

## Figures and Tables

**Figure 1 ijerph-18-05452-f001:**
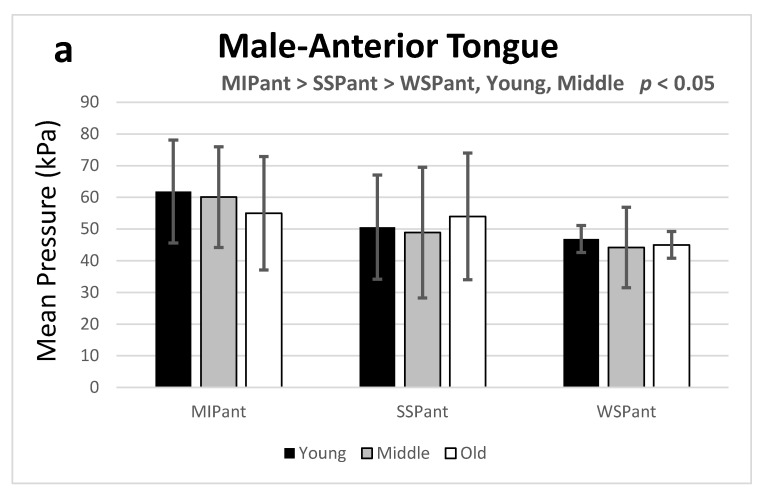
Maximum isometric pressure (MIP) and swallowing pressure (SP) during all tasks are shown for both sexes, with age group as the independent variable on the *x*-axis. Error bars throughout the results section represent standard deviation (±1 SD). Abbreviations: ant: anterior tongue; post: posterior tongue; MIP: maximum isometric pressure; SSP: saliva swallowing pressure; WSP: water swallowing pressure; ant: anterior of tongue; post: posterior of tongue.

**Table 1 ijerph-18-05452-t001:** Means and standard deviations for measures across age decades (N = 150).

Measure	Age (Years)	Mean ± SD	95% CI		Min	Max	Statistic	*p*	Effect Size (η^2^) ^b^
			Lower	Upper					
Maximum isometric pressure (MIP)									
MIP_ant_	20–29	57.91	54.13	61.68	17	92	F_4,145_ = 1.92	0.111	0.05
		(15.48)							
	30–39	58.56	52.92	64.19	34	87			
		(13.65)							
	40–49	53.13	47.93	58.33	26	93			
		(13.92)							
	50–59	54.67	49.46	59.86	30	69			
		(10.46)							
	60+	47.00	41.60	52.39	37	58			
		(7.54)							
MIP_post_ ^a^	20–29	56.15	53.25	59.03	32	82	F_4,145_ = 4.18	0.003	0.10
		(11.85)							
	30–39	55.40	50.20	60.59	25	80			
		(12.57)							
	40–49	49.47	44.54	54.38	25	79			
		(13.17)							
	50–59	51.61	47.33	55.88	32	67			
		(8.58)							
	60+	42.50	36.27	48.72	30	57			
		(8.70)							
Saliva swallowing pressure (SSP)									
SSP_ant_	20–29	49.61	45.97	53.25	13	75	F_4,145_ = 1.34	0.259	0.04
		(14.92)							
	30–39	47.56	40.30	54.81	21	75			
		(17.57)							
	40–49	47.73	41.90	53.56	19	86			
		(15.61)							
	50–59	48.00	41.64	54.35	17	62			
		(12.78)							
	60+	37.70	28.00	47.39	11	63			
		(13.55)							
SSP_post_	20–29	49.76	45.99	53.52	8	74	F_4,145_ = 1.01	0.404	0.03
		(15.43)							
	30–39	49.20	42.06	56.33	25	82			
		(17.28)							
	40–49	47.00	41.45	52.54	20	74			
		(15.86)							
	50–59	46.61	41.54	51.67	25	64			
		(10.18)							
	60+	40.30	32.03	48.56	22	55			
		(11.55)							
Water swallowing pressure (WSP)									
WSP_ant_	20–29	46.76	42.68	50.83	13	74	F_4,145_ = 1.09	0.364	0.03
		(16.69)							
	30–39	45.88	38.60	53.15	14	84			
		(17.62)							
	40–49	41.63	35.13	48.13	9	77			
		(17.41)							
	50–59	47.83	41.50	54.15	15	65			
		(12.71)							
	60+	38.40	32.38	44.41	21	48			
		(8.40)							
WSP_post_	20–29	44.48	40.76	48.19	6	66	F_4,145_ = 1.64	0.166	0.04
		(15.22)							
	30–39	45.68	38.45	52.90	14	82			
		(17.49)							
	40–49	39.50	33.26	45.73	12	84			
		(16.70)							
	50–59	47.00	41.52	52.47	26	66			
		(11.01)							
	60+	35.40	27.43	43.36	20	56			
		(11.13)							

^a^ Post hoc analysis indicated that participants aged 20–39 years had higher pressure than those aged >60 years; ^b^ η^2^, effect size; Abbreviations: CI: confidence interval; Min: minimum; Max: maximum; ant: anterior of tongue; post: posterior of tongue.

**Table 2 ijerph-18-05452-t002:** Differences in measures between the sexes (N = 150).

Measure	Sex	Mean ± SD	95% CI		Statistic	*p*	Effect Size (η^2^)
			Lower	Upper			
Maximum isometric pressure (MIP)							
MIP_ant_	Male	61.20	56.64	65.75	F_1,148_ = 10.72	0.001	0.07
		(15.85)					
	Female	53.40	50.92	55.87			
		(12.53)					
MIP_post_	Male	55.67	52.04	59.30	F_1,148_ = 2.93	0.089	0.02
		(12.62)					
	Female	52.05	49.69	54.40			
		(11.94)					
Saliva swallowing pressure (SSP)							
SSP_ant_	Male	50.35	45.55	55.13	F_1,148_ = 1.86	0.174	0.01
		(16.67)					
	Female	46.72	43.85	49.58			
		(14.51)					
SSP_post_	Male	47.53	43.14	51.91	F_1,148_ = 0.11	0.743	0.00
		(15.25)					
	Female	48.39	45.46	51.31			
		(14.81)					
Water swallowing pressure (WSP)							
WSP_ant_	Male	46.20	41.08	51.32	F_1,148_ = 0.30	0.585	0.00
		(17.83)					
	Female	44.65	41.60	47.70			
		(15.43)					
WSP_post_	Male	44.00	39.61	48.39	F_1,148_ = 0.12	0.733	0.00
		(15.27)					
	Female	43.08	40.01	46.15			
		(15.56)					

Abbreviations: CI: confidence interval; ant: anterior of tongue; post: posterior of tongue; η^2^, effect size.

**Table 3 ijerph-18-05452-t003:** Differences in parameters according to the areas of tongue.

Variable	Anterior Tongue M (SD)	Posterior Tongue M (SD)	*t*	*p*	Effect Size (η^2^)
MIP	55.95 (14.13)	53.23 (12.24)	3.16	0.077	0.01
SSP	47.91 (15.29)	48.11 (14.91)	0.01	0.909	0.00
WSP	45.16 (16.21)	43.38 (15.42)	0.95	0.331	0.00

Abbreviations: M: mean; MIP: maximum isometric pressure; SD: standard deviation; SSP: saliva swallowing pressure; WSP: water swallowing pressure; *t*: paired-*t* value; η^2^: effect size.

**Table 4 ijerph-18-05452-t004:** Tongue pressure generated by different tasks.

Variable	Mean (SD)	95% CI	F	*p*	Effect Size	(η^2^)	HSD
Anterior tongue							
MIP	55.95 (14.13)	53.66–58.22	20.29	<0.001	0.08		MIP > SSP > WSP
SSP	47.91 (15.29)	45.43–50.37					
WSP	45.16 (16.21)	42.54–47.77					
Posterior tongue							
MIP	53.23 (12.24)	51.25–55.20	17.91	<0.001	0.10		MIP > SSP > WSP
SSP	48.11 (14.91)	45.70–50.51					
WSP	43.38 (15.42)	40.89–45.86					

Abbreviations: CI, confidence interval; η^2^, effect size; HSD: honestly significant difference; MIP: maximum isometric pressure; SD, standard deviation; SSP: saliva swallowing pressure; WSP: water swallowing pressure.

## Data Availability

The datasets used and/or analyzed during the present study are available from the corresponding author on reasonable request.

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
