# Peer review of "Age and Sex-Related Differences in the Tongue Pressure Generated during Maximum Isometric and Swallowing Tasks by Healthy Chinese Adults"

_ijerph, 2021, doi:10.3390/ijerph18105452_

Round 1

Reviewer 1 Report

The title is too long, I advise authors to make it shorter and more INTERESTING 

For example :The Tongue  Pressure Generated During Maximum Isometric and Swallowing Tasks : A Prospective observational study 

Better specify aim, methods , results, conclusion  Abstract:

Aim  The aims of this prospective observational study were to investigate age, sex, and the 14 factors related to the tongue pressure generated. A correlational research design was used. Methods A total 15 of 150 Chinese people who had a normal swallowing condition were enrolled by convenience sam- 16 pling. Pressure was measured for each participant during maximum isometric press tasks, as well 17 as for saliva and water swallows (5 ml) at the anterior and posterior tongue.

Results :The results illustrated 18 that age has an impact on anterior tongue pressure (r = -0.22), posterior tongue pressure (r = -0.26); 19 however, it does not have an impact on the swallowing pressure (SP) of the tongue. Sex differences 20 were noted; males demonstrated a greater strength of the anterior tongue. There was a significant 21 correlation between BMI and the maximum isometric pressure of the anterior tongue (MIPant). The 22 pressures between anterior and posterior tongue were not significantly different in the maximum 23 isometric or swallowing tasks. There were significant differences among the maximum isometric 24 pressure (MIP), saliva swallowing pressure, and water swallowing pressure. The MIP generated 25 was greater than the pressure in the swallowing tasks for the younger groups of both sexes. Given 26 that MIP is known to decline with age and that greater increase is seen in various training programs, 27 adult people especially female sex are suggested to conduct lingual strength training, and may halt 28 or prevent presbyphagia.

Conclusion The study supplement the exploration of age-and-sex related differences 29 and the interaction of sex and age in tongue pressure.

Author Response

Manuscript Reference Number: ijerph-1096836

Title: Age and Sex-related Differences in the Tongue Pressure Generated During Maximum Isometric and Swallowing Tasks by Healthy Chinese Adults

Journal: IJERPH

Reviewer 1 comments

Response to Reviewer 1

1.          The title is too long, I advise authors to make it shorter and more interesting.

1.          We appreciate the important points that Reviewer 1 raised regarding the title. To address this point and the availability of data on the Chinese population, the title of the manuscript has been changed to “Age and Sex-related Differences in the Tongue Pressure Generated During Maximum Isometric and Swallowing Tasks by Healthy Chinese Adults”.

2.          Better specify aim, methods , results, conclusion in the Abstract.

2.          We are very appreciative the suggestion that Reviewer 1 raised regarding the abstract. The format of the abstract is written according to this journal guide.

Reviewer 2 Report

Congratulations for this work.

Author Response

Manuscript Reference Number: ijerph-1096836

Title: Age and Sex-related Differences in the Tongue Pressure Generated During Maximum Isometric and Swallowing Tasks by Healthy Chinese Adults

Journal: IJERPH

Reviewer 2 comments

Response to Reviewer 2

1.          Congratulations for this work.

1.          We appreciate the Reviewer’s encouragement.

Reviewer 3 Report

The manuscript reports a secondary analysis of the results of the study by Wu et al published on this journal in 2020. Age, gender, BMI, and tasks differences were investigated on tongue pressure measures. The topic is not novel as other studies in literature have investigated similar data. On the other hand, the included outcomes seems to vary across different countries, thus, the availability of data on the Chinese population may be useful in clinical practice. The paper is well written, the study methodology is linear and clear. I have few minor comments:

  • I would increase the focus on the fact that the study reports data on the Chinese population both in the title and in the abstract
  • Page 1, lines 26-28: I would remove the sentence "Given adult people especially female sex are suggested to conduct lingual strength training and may halt or prevent presbyphagia" from the abstract as this is an hypotesis and not supported from the results of the present study
  • page 2 line 67 "conducted with large, stratified samples" I would not define 150 subjects divided into 5 age groups as a large sample
  • The fact that the 60+ age group only contains 10 subjects is a major limit and should be more stressed in the limits of the study
  • MIP: was the IOPI calibrated? How often?
  • SP: the procedure to acquire WSP measures should be describe in details
  • the methods used to acquire data on the BMI should be described
  • page 9 lines 313-322 the whole paragraph should be nuanced as the potential for the efficacy of a training program on healthy older subjects to prevent presbyphagia is only speculative.

Author Response

Manuscript Reference Number: ijerph-1096836

Title: Age and Sex-related Differences in the Tongue Pressure Generated During Maximum Isometric and Swallowing Tasks by Healthy Chinese Adults

Journal: IJERPH

Reviewer 3 comments

Response to Reviewer 3

1.          The manuscript reports a secondary analysis of the results of the study by Wu et al published on this journal in 2020. Age, gender, BMI, and tasks differences were investigated on tongue pressure measures. The topic is not novel as other studies in literature have investigated similar data. On the other hand, the included outcomes seems to vary across different countries, thus, the availability of data on the Chinese population may be useful in clinical practice. The paper is well written, the study methodology is linear and clear. I have few minor comments:

1.          We appreciate the Reviewer’s encouragement.

2.          I would increase the focus on the fact that the study reports data on the Chinese population both in the title and in the abstract

2.          To address this point, we have modified the title and abstract.

The title of the manuscript has been changed to “Age and Sex-related Differences in the Tongue Pressure Generated During Maximum Isometric and Swallowing Tasks by Healthy Chinese Adults”. Abstract : “The aims of this prospective observational study were to investigate age, sex, and the factors re-lated to the tongue pressure generated. A correlational research design was used. A total of 150 Chinese people who had a normal swallowing condition were enrolled by convenience sam-pling. Pressure was measured for each participant during maximum isometric press tasks, as well as for saliva and water swallows (5 ml) at the anterior and posterior tongue. The results il-lustrated that age has an impact on anterior tongue pressure (r = -0.22), posterior tongue pressure (r = -0.26); however, it does not have an impact on the swallowing pressure (SP) of the tongue. Sex differences were noted; males demonstrated a greater strength of the anterior tongue. There was a significant correlation between BMI and the maximum isometric pressure of the anterior tongue (MIPant). The pressures between anterior and posterior tongue were not significantly dif-ferent in the maximum isometric or swallowing tasks. There were significant differences among the maximum isometric pressure (MIP), saliva swallowing pressure, and water swallowing pressure. The MIP generated was greater than the pressure in the swallowing tasks for the younger groups of both sexes. The study supplement the exploration of age-and-sex related dif-ferences and the interaction of sex and age in tongue pressure.

3.          Page 1, lines 26-28: I would remove the sentence "Given adult people especially female sex are suggested to conduct lingual strength training and may halt or prevent presbyphagia" from the abstract as this is an hypotesis and not supported from the results of the present study

3.          In the revised version, we have removed this sentence.

4.          page 2 line 67 "conducted with large, stratified samples" I would not define 150 subjects divided into 5 age groups as a large sample

4.          We have deleted this word. “Furthermore, additional research that examines differences in tongue pressure across the age continuum conducted with stratified samples is necessary for comparison with prior research studies to investigate the similarities and differences.” (p.1)

5.          The fact that the 60+ age group only contains 10 subjects is a major limit and should be more stressed in the limits of the study

5.          We appreciate the important point that the Reviewer 3 raised regarding the 60+ age group. To address this point, we have modified in some sections:

There were few participants older than 60 years (n = 10) and they were put into one group for practical reasons.” (p.3)

It might originate from the fewer participants in the older adult group and thus limit the outward differences among groups.” (p.9)

Firstly, the presence of fewer subjects in the oldest age category, and fewer males than females may elicit selection bias. Therefore, a study involving a sufficiently diverse group of participants to capture age- and sex-related changes is warranted to verify the results of the present study.” (p.9)

6.          MIP: was the IOPI calibrated? How often?

6.          We have added this point to the section of Maximum isometric pressure: “To ensure accurate measurement, calibration of the IOPI was conducted once a week ac-cording to the IOPI manual.” (p.2)

7.          SP: the procedure to acquire WSP measures should be describe in details

7.          To address this point, we have revised our statement in the section of swallowing pressure: “SP was defined as the non-effortful swallowing pressure across three consecutive trials for saliva swallowing and thin liquid (commercially bottled water) swallowing. The bulb was placed in the specified anterior or posterior position used in the earlier MIP measurements. The bolus was offered by the investigator in a cup (5 mL) and swallowed by the participant with the tongue bulb in the lingual regions in a comfortable manner.” (p.2)

8.          the methods used to acquire data on the BMI should be described

8.          In agreement with this point, we have added the following to the BMI-related differences section: “The BMI parameter was calculated as participants body weight divided by their body height squared (kg/m2) and dichotomized into a categorical variable (normal: 18.5–24.0 kg/m2; abnormal: < 18.5 kg/m2 or > 24.0 kg/m2).” (p.5)

9.          page 9 lines 313-322 the whole paragraph should be nuanced as the potential for the efficacy of a training program on healthy older subjects to prevent presbyphagia is only speculative.

9.          We appreciate the important point that Reviewer 3 raised regarding the discussion. We have deleted the whole paragraph. (p.9)

This manuscript is a resubmission of an earlier submission. The following is a list of the peer review reports and author responses from that submission.

Round 1

Reviewer 1 Report

1)The title needs of  modification : reduce repetitions and make a more interesting title

Associations of Age, Sex, Body Mass Index, Pressure Generating Tasks, and Areas of the Tongue with the Tongue Pressures 3 Generated by Chinese People

2)Abstract: to specify aim, methods ,results, conclusion

3) 82-85 Methods :to specify the methods

4)Conclusions : The conclusions are not yet sufficient to corroborate the usefulness of this study

Author Response

Manuscript Reference Number: ijerph-1096836

Title: Investigation of Factors Influencing on the Tongue Pressure Generated by Adult People

Journal: IJERPH

Reviewer 1 comments

Response to Reviewer 1

1.          The title needs of modification : reduce repetitions and make a more interesting title

Associations of Age, Sex, Body Mass Index, Pressure Generating Tasks, and Areas of the Tongue with the Tongue Pressures 3 Generated by Chinese People

1.          We thank Reviewer 1 for the constructive comments. We have revised the article title to “Investigation of Factors Influencing on the Tongue Pressure Generated by Adult People” (lines 2-3).

2.          Abstract: to specify aim, methods ,results, conclusion

2.          The abstract has been revised two parts: “The aims of this prospective observational study were to investigate the factors related to the tongue pressure generated. A correlational research design was used. A total of 150 Chinese people who had a normal swallowing condition were enrolled by convenience sampling.” (lines 13-14). And “The MIP generated was greater than the pressure in the swallowing tasks for the younger groups of both sexes. The study supplement the exploration of age-and-sex related differences and the interaction of sex and age in tongue pressure.” (lines 26-27).

3.          82-85 Methods :to specify the methods

3.          We specify the methods “A correlational research design was conducted to investigate the associations among age, sex, BMI, MIP, and swallowing pressure (SP) generated by the anterior and posterior tongue.” (lines 82-84).

4.          Conclusions : The conclusions are not yet sufficient to corroborate the usefulness of this study

4.          We appreciate the important points that Reviewer 1 raised with regard to the conclusion. It has been revised as follows: “There has not been an investigation on MIP, SSP, and WSP data of the tongue in a Chinese population that varied across age and sex. This study provides the first Chinese data on tongue pressure generating tasks. In the present study, advanced age and female sex appeared to be associated with lower MIP. There were no significant differences between the anterior and posterior tongue pressure for either MIP or SP. There are significant differences in tongue pressure among MIP, SSP, and WSP. The MIP is greater than the pressure generated by either swallowing task for the young groups of both sexes. The MIP, SSP, and WSP data of the tongue help in the comprehension of swallowing physiology, and will thus contribute to therapeutic planning for individuals with dysphagia. The current study expands the dataset describing factors related to tongue pressure measures, especially the interaction of sex and age in tongue pressure.” (lines 329-338).

Reviewer 2 Report

Dear Authors,

I have received your comments and I'm fully satisfied.

In my opinion the manuscript in the revised form is suitable for publication in IJERPH.

Kind regards

Author Response

Manuscript Reference Number: ijerph-1096836

Title: Investigation of Factors Influencing on the Tongue Pressure Generated by Adult People

Journal: IJERPH

Reviewer 2 comments

Response to Reviewer 2

1.          I have received your comments and I'm fully satisfied.

In my opinion the manuscript in the revised form is suitable for publication in IJERPH.

1.          We appreciate the Reviewer 2 encouragement.

Reviewer 3 Report

Comments for Manuscript ID: ijerph-1163846

Authors demonstrated that maximum isometric pressure (MIP) of anterior tongue pressure declined with age and higher that in male comparing with female. The tongue pressure using during swallowing was lower than MIP, regardless sex and age. Although the examinations were well conducted, there was few new findings. The domestic journal in China or Taiwan would be suitable for publication for this manuscript.

p.5, l.172-179: BMI was divided into two groups, normal and abnormal. Why did authors divide participants into three groups, i.e., lean, normal and obese? Given that the BMI correlated with MIP, MIP of lean participants might be higher than that of obese participants.

p.8, l.231-232: Authors argued that age-related reduction in MIP for anterior and posterior region of tongue were observed. However, only MIP of posterior tongue indicated statistical significance among age groups (Table 1). The discrepancy should be explained. In addition, if authors would like to argue the correlation between age and MIP, adjustment with confounders should be needed. In this line, linear regression analysis would be useful to examine the correlation between MIP and parameters of participants, such as age, sex, and BMI.

p.9, l.304: “Hence, greater SP is used to compensate for the reduction”. An additional explanation should be required what authors explain by this sentence.

Author Response

Manuscript Reference Number: ijerph-1096836

Title: Investigation of Factors Influencing on the Tongue Pressure Generated by Adult People

Journal: IJERPH

Reviewer 3 comments

Response to Reviewer 3

1.          p.5, l.172-179: BMI was divided into two groups, normal and abnormal. Why did authors divide participants into three groups, i.e., lean, normal and obese? Given that the BMI correlated with MIP, MIP of lean participants might be higher than that of obese participants.

1.          We appreciate the important points that Reviewer 3 raise with regard to the BMI. “The BMI parameter was calculated as weight divided by height squared (kg/m2) and dichotomized into a categorical variable (normal: 18.5–24.0 kg/m2; abnormal: < 18.5 kg/m2 or > 24.0 kg/m2).” (lines 173-175). It was divided into two groups, normal and abnormal.

2.          p.8, l.231-232: Authors argued that age-related reduction in MIP for anterior and posterior region of tongue were observed. However, only MIP of posterior tongue indicated statistical significance among age groups (Table 1). The discrepancy should be explained. In addition, if authors would like to argue the correlation between age and MIP, adjustment with confounders should be needed. In this line, linear regression analysis would be useful to examine the correlation between MIP and parameters of participants, such as age, sex, and BMI.

2-1      Although only MIP of posterior tongue indicated statistical significance among age group (Table 1), the Pearson’s r correlation coefficients between age and tongue pressure variables revealed that age has an impact on the MIP of the tongue (anterior tongue pressure r = −0.22; posterior tongue pressure r = −0.26); (lines 153-157).

The results of linear regression analysis also supported the findings (lines 180-182).

Hence, “Age-related reductions in tongue pressure were clearly observed in MIP, both for the anterior and posterior tongue, and particularly for individuals aged ≥ 60 years.” (lines 234-235).

2-2      We thank Reviewer 2 for the constructive comments with regard to the statistical analysis. We have added to the section of statistical analysis “In addition, linear regression was used to examine the correlation between MIP and parameters of participants.” (lines 139-141). And the section of results “Linear regression analysis between MIP and parameters of participants, such as age, sex, and BMI were presented in Appendix 3. Age-related reductions in tongue pressure were significant in MIPant and MIPpost.” (lines 180-182).

Appendix 3. Linear regression analysis between MIP and parameters of participants

Variable

MIPant

MIPpost

B

SE

t

p

B

SE

t

p

Age

-0.18

0.07

-2.46

.015

-0.20

0.07

-3.12

.002

Sex

-5.78

2.44

-2.37

.019

-2.47

2.16

-1.14

.255

BMI

0.56

0.29

1.93

.055

0.15

0.25

0.58

.564

3.          p.9, l.304: “Hence, greater SP is used to compensate for the reduction”. An additional explanation should be required what authors explain by this sentence.

3.          The reasoning in support of the findings has been added to the text “The reason for this effect may be the overall decrease in orofacial muscle strength and tone in older adults. Hence, older adults need to use greater SP to compensate for the reduction.” (lines 316-318).

Round 2

Reviewer 1 Report

change the title: the title does not reflect the article

abstract: follow and to specify  the model for purpose results conclusions 

Reviewer 3 Report

I appreciate that the authors replied to reviewer’s comments. However, the manuscript has a fundamental issue not to be able to address, which I indicated at first round. The manuscript did not show a novel finding regarding factors influencing on the tongue pressure.